# Alcohol-Associated Liver Disease Outcomes: Critical Mechanisms of Liver Injury Progression

**DOI:** 10.3390/biom14040404

**Published:** 2024-03-27

**Authors:** Natalia A. Osna, Irina Tikhanovich, Martí Ortega-Ribera, Sebastian Mueller, Chaowen Zheng, Johannes Mueller, Siyuan Li, Sadatsugu Sakane, Raquel Carvalho Gontijo Weber, Hyun Young Kim, Wonseok Lee, Souradipta Ganguly, Yusuke Kimura, Xiao Liu, Debanjan Dhar, Karin Diggle, David A. Brenner, Tatiana Kisseleva, Neha Attal, Iain H. McKillop, Shilpa Chokshi, Ram Mahato, Karuna Rasineni, Gyongyi Szabo, Kusum K. Kharbanda

**Affiliations:** 1Department of Pharmacology and Experimental Neuroscience, University of Nebraska Medical Center, Omaha, NE 68106, USA; 2Department of Internal Medicine, University of Nebraska Medical Center, Omaha, NE 68106, USA; 3Department of Internal Medicine, University of Kansas Medical Center, Kansas City, KS 66160, USA; itikhanovich@kumc.edu; 4Department of Medicine, Division of Gastroenterology, Beth Israel Deaconess Medical Center, Harvard Medical School, Boston, MA 02115, USA; mortegar@bidmc.harvard.edu (M.O.-R.); gszabo1@bidmc.harvard.edu (G.S.); 5Center for Alcohol Research, University of Heidelberg, 69120 Heidelberg, Germany; sebastian.mueller@urz.uni-heidelberg.de (S.M.); chaowenzheng5@gmail.com (C.Z.); joh-mueller@gmx.de (J.M.); siyuanli.med@gmail.com (S.L.); 6Viscera AG Bauchmedizin, 83011 Bern, Switzerland; 7Department of Medicine, University of California San Diego, La Jolla, CA 92093, USA; ssakane@health.ucsd.edu (S.S.); rcgweber@health.ucsd.edu (R.C.G.W.); hyk047@health.ucsd.edu (H.Y.K.); wol014@health.ucsd.edu (W.L.); soganguly@health.ucsd.edu (S.G.); y1kimura@health.ucsd.edu (Y.K.); xil094@health.ucsd.edu (X.L.); ddhar@health.ucsd.edu (D.D.); kdiggle@health.ucsd.edu (K.D.); dbrenner@sbpdiscovery.org (D.A.B.); 8Department of Surgery, University of California San Diego, La Jolla, CA 92093, USA; tkisseleva@health.ucsd.edu; 9Sanford Burnham Prebys Medical Discovery Institute, La Jolla, CA 92037, USA; 10Department of Surgery, Atrium Health Carolinas Medical Center, Charlotte, NC 28203, USA; neha.attal@atriumhealth.org (N.A.); iain.mckillop@atriumhealth.org (I.H.M.); 11The Roger Williams Institute of Hepatology, Foundation for Liver Research, London SE59NT, UK; s.chokshi@researchinliver.org.uk; 12School of Microbial Sciences, King’s College, London SE59NT, UK; 13Department of Pharmaceutical Science, College of Pharmacy, University of Nebraska Medical Center, Omaha, NE 68106, USA; ram.mahato@unmc.edu; 14Department of Biochemistry and Molecular Biology, University of Nebraska Medical Center, Omaha, NE 68106, USA; karuna.rasineni@unmc.edu; 15Research Service, Veterans Affairs Nebraska-Western Iowa Health Care System, Omaha, NE 68105, USA

**Keywords:** alcohol-associated liver disease, epigenetics, cell death, hemolysis, MetAld, hepatic stellate cells, fibrosis, fatty acid binding protein 4, hepatocellular carcinoma, models

## Abstract

Alcohol-associated liver disease (ALD) is a substantial cause of morbidity and mortality worldwide and represents a spectrum of liver injury beginning with hepatic steatosis (fatty liver) progressing to inflammation and culminating in cirrhosis. Multiple factors contribute to ALD progression and disease severity. Here, we overview several crucial mechanisms related to ALD end-stage outcome development, such as epigenetic changes, cell death, hemolysis, hepatic stellate cells activation, and hepatic fatty acid binding protein 4. Additionally, in this review, we also present two clinically relevant models using human precision-cut liver slices and hepatic organoids to examine ALD pathogenesis and progression.

## 1. Introduction

Alcohol-associated liver disease (ALD) is a substantial cause of morbidity and mortality worldwide that progresses through various stages starting with fatty liver (steatosis) to inflammation and fibrosis–cirrhosis, factors that increase the risk of developing hepatocellular carcinoma (HCC) [1]. Nearly half of the deaths from liver cirrhosis are attributable to alcohol use disorders (AUDs) [2]. In addition to serving as a primary cause of liver injury, alcohol also accelerates fibrosis progression in patients with metabolic syndrome or viral hepatitis (hepatitis B and/or C infection), thereby negatively affecting outcomes of patients with chronic liver diseases arising from different etiologies [3]. Multiple factors contribute to ALD progression, and these cannot be fully summarized in one review article. Here, we will focus on some of the mechanisms that play an important role in advancing liver injury triggered by alcohol toward end-stage liver disease. In this regard, we start by reviewing epigenetic changes associated with ALD that promote liver disease development and progression. Next, we present alcohol-induced cell death as a central mechanism for hepatotoxicity and disease severity in acute-on-chronic liver failure. We also overview hemolysis, a less investigated ALD-related phenomenon, related to alcohol-associated mortality. Since the status of hepatic stellate cells (HSCs) is fundamental for the initiation of fibrosis/cirrhosis, we then focus on mechanisms regulating HSC activation/inactivation in subjects with metabolic dysfunction-associated steatotic liver disease (MASLD) and excessive alcohol intake (MetALD). Next, we overview the role of hepatic fatty acid binding protein 4 (FABP4) in the development of alcohol-dependent fatty liver and liver tumor development. Finally, we present two clinically relevant models using human precision-cut liver slices (PCLS) and hepatic organoids to examine ALD pathogenesis and progression. 

## 2. Epigenetics in ALD Progression

ALD development and progression often involve gene activation and suppression without changes in DNA sequence [4,5,6,7,8,9,10]. These changes are mediated by histone and DNA modifications that are part of the spectrum of epigenetic modifications. Major modifications that are affected by alcohol are acetylation and methylation (Figure 1). Ethanol has been proven to have several deleterious effects on acetylation and methylation, such as increased gene-selective levels of histone H3 acetylation, increased levels of enzymes mediating histone acetylation, and a generalized increase in DNA methylation.

These epigenetic modifications affect all liver cell types. Some of them, such as epigenetic changes in hepatocytes, are likely due to the direct effect of alcohol metabolism or alcohol-induced oxidative stress that results in altered gene expression of epigenetic regulators. Others, such as epigenetic changes in macrophages, are likely indirect and are due to changes in the liver microenvironment or gut–liver crosstalk.

### 2.1. Alcohol-Induced Hepatocyte Epigenetic Changes 

Alcohol metabolism can directly impact histone acetylation in hepatocytes. The global increase in histone H3 at lysine 9 acetylation (H3AcK9) initially has been observed in primary rat hepatocytes exposed to alcohol in vitro [11]. Apart from global changes in histone acetylation level, several acetylation changes were reported at the gene-selective level. Global acetylation changes are likely mediated by an altered balance of histone acetyltransferases (HATs) and histone deacetylation (HDAC) enzymes [5,10,11,12]. Several studies aimed to target alcohol-induced imbalance in acetylation enzymes by HATs or HDAC-targeted drugs; however, the results were not conclusive [9,13,14,15,16,17,18,19]. These data suggest that selective targeting of acetyltransferases and deacetylases is necessary. One of the attractive targets is a deacetylase SIRT6; studies have shown that it protects the liver from alcohol-induced tissue injury by reducing oxidative stress in mice by regulating metallothionein 1 and 2 expression [17].

Reduced S-adenosylmethionine levels contribute to altered histone methylation status [8,20,21,22,23,24]. These changes are exacerbated by alcohol-induced alterations in histone lysine and arginine methyltransferases and demethylation enzymes [20,25,26,27]. Histone arginine methyltransferases PRMTs and demethylase JMJD6 in hepatocytes control hepatic oxidative stress and liver injury as well as hepatocyte differentiation and HCC development in alcohol-exposed livers [25,28,29]. One of the mechanisms described involves PRMT1-dependent histone arginine methylation of the *Hnf4a* gene promoter. Alcohol-induced decrease in PRMT activity and an increase in JMJD6 contribute to HNF4α loss in ALD and promote ALD development [20,25,28,29].

Histone lysine demethylases KDM5B and KDM5C were shown to promote hepatocyte dedifferentiation and reduce HNF4α expression in response to alcohol [30]. Collective epigenetic suppression of HNF4α-dependent transcription was shown to be the main driver of liver function loss in ALD [31]. Thus, epigenetic enzyme targeting is an attractive target for restoring hepatocyte function in ALD.

Transcriptional co-regulators play key roles in the epigenetic regulation of gene expression. Nuclear receptor corepressor 1 (NCoR1) is a corepressor of the epigenetic regulation of gene transcription that has important functions in metabolism and inflammation. Hepatocyte NCoR1 plays distinct roles in controlling liver inflammation and steatosis by promoting fatty acid oxidation and reducing CCL2 gene expression [32].

The 5-hydroxymethylcytosine (5hmc) is a newly identified epigenetic modification thought to be regulated by the TET family of proteins. Chronic ethanol-mediated hepatocyte apoptosis is linked to decreased TET1 and 5-hydroxymethylcytosine formation, which could contribute to ALD development [33].

The global increase in DNA methylation also contributes to ALD, as evidenced by the reduced susceptibility of DNA methyltransferase 1 hypomorphic (Dnmt1N/+) mice to hepatic steatosis upon feeding a liquid alcohol diet [34].

A recent study using scATAC-seq analysis confirmed that alcohol-induced epigenetic changes alter the transcriptional program in hepatocytes of alcohol-exposed mice. Moreover, alcohol-induced epigenetic changes in hepatocytes persist after alcohol cessation and contribute to poor disease resolution after alcohol cessation [35].

### 2.2. Immune Cell Epigenetic Changes

Alcohol-induced changes in histone modification enzyme expression and activity affect multiple innate and adaptive immunity cell populations [36,37,38,39,40,41,42]. Alcohol-induced epigenetic changes were described for several types of immune cells (e.g., granulocytes, macrophages, and T-lymphocytes) and are thought to promote exaggerated inflammatory responses in ALD. Alcohol exposure promotes altered macrophage polarization and sensitivity to endotoxin stimulation, skewed cytokine production profile, altered phagocytic activity, and reduced capacity to present antigen [39,40]. Alcohol alters macrophage polarization in the liver. Some of these effects could be due to macrophage replacement induced by alcohol [43,44]; however, the contribution of epigenetic changes in both resident Kupffer cells and infiltrating monocyte-derived macrophages have been reported [35].

Alcohol-induced changes in histone arginine methylation enzymes, PRMT1, and PRMT6 in myeloid cells have been extensively studied. Like the effect in hepatocytes, alcohol reduces PRMT activity in myeloid cells. In addition, the alteration of arginine methyltransferases substrate specificity induced by alcohol promotes alcohol-induced macrophage dysfunction that contributes to the development of liver fibrosis and alcohol-associated HCC [45,46,47,48,49,50].

The roles of other histone modification enzymes and DNA methylation in immune cell phenotype were not specifically explored in ALD. However, the data from other disease models suggest that they could be implicated in altered cytokine production in macrophages and dendritic cells as well as altered CD8 + and CD4 + T cell phenotypes that promote liver injury and enhance alcohol-induced liver disease progression [37,40]. 

### 2.3. Hepatic Stellate Cell (HSC) Epigenetic Changes

Hepatocellular damage and inflammation stimulate the transdifferentiation of resident perisinusoidal HSC into α-smooth muscle actin (αSMA)-positive myofibroblasts. These so-called “activated” HSCs (aHSC) are the major hepatocellular source of fibrotic ECM proteins and promote the deposition of fibrotic ECM in chronic liver disease. Hepatic stellate cell activation involves genome-wide remodeling of the DNA methylation and histone modification landscapes. G9a and DNMT1 are some of the recently discovered drivers of HSC activation [51,52,53,54,55,56].

Alcohol can directly stimulate epigenetic modifications in hepatic stellate cells [57]. HSCs respond to ethanol exposure by increasing profibrogenic and ECM gene expression, which correlates with the ethanol-induced altered expression of multiple epigenetic regulators, including MLL1, a histone 3 lysine 4 (H3K4) methyltransferase. MLL1-mediated H3K4me3 methylation is implicated in HSC activation. Several other H3K4 methyltransferases and demethylases were altered by alcohol, including H3K4 demethylases, KDM5A, and KDM5B, as well as several enzymes regulating H3K9 methylation and H3K27 methylation.

Recent studies identified KDM5B as a key regulator of HSC activation and fibrosis development in alcohol-exposed females in vitro and in vivo. Alcohol-induced KDM5B activation was shown to promote AhR pathway suppression in hepatic stellate cells which, in turn, results in fibrogenic gene expression in HSCs [58].

### 2.4. Liver Sinusoidal Endothelial Cell (LSEC) Epigenetic Changes

Alcohol impacts the epigenetic landscape of LSECs more than any other liver non-parenchymal cell population [35]. However, the mechanism and the role of these epigenetic changes in specific functions of LSECs are not fully understood. LSECs are important in liver homeostasis. They also modulate the response to liver injury by initiating hepatocyte regeneration [59,60]. In addition, LSECs modulate liver fibrosis development in a zone-specific way [61,62]. Other models of liver disease have indicated the important role of epigenetic regulation in LSECs in the development of liver disease. Suppressed histone H3K18 acetylation is implicated in arsenic-induced liver fibrosis by modulating LSEC differentiation [63]. Mechanotransduction-induced glycolysis epigenetically regulates the LSEC CXCL1-signaling program [64]. Endothelial p300 interaction with NF-kappa B and BRD4 increases CCL2 expression, leading to macrophage accumulation, portal hypertension, and liver fibrosis [7]. 

In alcohol-associated hepatitis, sinusoidal endothelial cells are identified as an important source of CXCL expression, which is driven by epigenetic super-enhancer activation downstream of TNFα signaling [65]. In addition, LSECs can metabolize alcohol, which results in altered acetylation of proteins [66]; thus, LSEC-mediated alcohol metabolism could also lead to alcohol-dependent epigenetic changes similar to those in hepatocytes.

## 3. Acute-on-Chronic Liver Failure

Acute-on-chronic liver failure (ACLF) is a complex syndrome arising in patients with advanced/end-stage chronic liver disease after an acute decompensation event that leads to liver failure [67]. ACLF is characterized by systemic inflammation, liver and multi-organ failure, and reduced survival [68]. Multiple studies found that alcohol is the most frequent precipitating factor of ACLF [69,70]. In the context of liver diseases, alcohol is known to induce many, if not all, types of cell death [71], potentially playing a central role in the progression of liver injury in ACLF.

### 3.1. ACLF and Cell Death

Both clinical and translational studies have highlighted hepatocyte death as a key feature in ACLF pathophysiology [72]. Damage-associated molecular patterns are immunogenic molecules released by dying or dead cells that contribute to inflammation and other aspects of ACLF pathophysiology. While some authors postulate that apoptosis is the predominant mode of cell death during ACLF [73,74], evidence supports a major contribution of non-apoptotic pathways, including necroptosis and pyroptosis [75,76].

Adabayo and colleagues [74] showed that the circulating levels of genomic DNA and the epithelial cell death marker, caspase-cleaved cytokeratin-18 (CK-18), were higher in their cohort of ACLF patients (predominantly induced by alcohol) compared to their non-ACLF controls. In a similar study from HBV-associated ACLF, the authors reported submissive necrosis as opposed to apoptosis [77], highlighting the relevance of the etiological factor in regulating the predominant cell death pathway during ACLF. PTEN-induced kinase 1 (PINK1) has recently been described as a link regulator for mitochondrial function and apoptosis in the context of ACLF [73]. The authors found that livers from ACLF patients showed decreased PINK levels and increased cleaved caspase 3 levels. Adenoviral overexpression of PINK1 in a murine ACLF model induced by chronic CCl_4_ and acute LPS + D-Gal inhibited apoptosis through the activation of the mTORC2/p-AKT pathway.

On the other hand, Kondo and colleagues [75] studied the role of RIPK3 as a landmark marker for necroptosis in ACLF. They showed that circulating RIPK3 levels were higher in patients with ACLF compared to non-ACLF patients. Increased levels of RIPK1 and RIPK3 were confirmed in the liver of patients with ACLF and a murine model of ACLF consisting of bile duct ligated mice receiving LPS. Interestingly, patients with ACLF due to alcohol etiology exhibited higher RIPK3 levels in serum than ACLF patients due to other etiologies. They also showed that RIPK1 inhibition by RIPA56 prevents RIPK1/RIPK3-mediated necroptosis in a CCl_4_ + GalN mouse model without changes in apoptotic markers. Moreover, Khanam et al. [78] showed that necrosis can be prevented by inhibiting the inflammatory process triggered by the CXCR1/2 receptors, which are highly expressed in peripheral blood mononuclear cells from ACLF patients. Even though pyroptosis has not been addressed in the context of ACLF, there is little evidence of the activation of this cell death pathway in acute liver failure (ALF) [76]. Li et al. [79] described that pyroptosis contributes to macrophage recruitment and inflammatory signaling in an ALF model of LPS + GalN. The inhibition of pyroptosis using gasdermin D knockout mice [79], limonin [80], or the TNFα inhibitor CC-5013 [81] ameliorated ALF.

### 3.2. The Role of Alcohol in Cell Death and ACLF

In the specific context of alcohol and ACLF, a mouse model of combined CCl_4_ exposure with chronic alcohol exposure by intragastric alcohol feeding [82], weekly binges [83], or the addition of alcohol in the drinking water [84] induced ACLF. These chronic models generally recapitulated liver injury, including fibrosis, steatohepatitis, and inflammation. Moreover, alcohol further induced ALT and necrosis in the liver, hypoxia, and histone modification.

A recent work presented at the American Association for the Study of Liver Diseases annual meeting [85] described the role of acute alcohol as the precipitating factor of ACLF in a novel murine model of cholestatic fibrosis and a single alcohol binge [85]. Mice undergoing ACLF showed increased cell death, as shown by increased circulating levels of ALT and cytokeratin 18. Specifically, ACLF mice exhibited no change in apoptotic-related markers (caspase-cleaved CK18, cleaved caspase 3) and an increase in RIPK3 and NLRP3, gasdermin D, IL-1b, and IL-18 when compared to fibrotic mice receiving water, validating the major role of necroptosis in the disease pathobiology and characterizing, for the first time, the role of pyroptosis in alcohol-induced ACLF. In this model, the authors also explored the mechanistic role of G-CSF as a therapeutic option for ACLF [86]. At the molecular level, Ortega-Ribera and colleagues showed that G-CSF induces neutrophil infiltration/activation, calprotectin (S100a8 and S100a9 heterodimer) release, oxidative stress, and inflammasome activation in both the liver and the brain. Moreover, they found increased neuroinflammation, type I interferon response, and extracellular matrix remodeling/fibrosis in the liver. Overall, the study concluded that G-CSF does not ameliorate ACLF pathophysiology in a murine model of alcohol-induced ACLF; contrarily, it increased liver damage and neuroinflammation.

Ferroptosis is a programmed cell death characterized by the iron-dependent accumulation of hydroperoxides, leading to ROS and lipid peroxidation. Alcohol has been described to induce iron accumulation through hepcidin signaling. Zhou and colleagues [87] found that intestinal-specific SIRT1 inhibition ameliorated hepatic inflammation and liver injury in a model of the chronic plus binge alcohol feeding model. Moreover, adipose-specific lipin-1 overexpression [88] under the influence of alcohol exhibited and exacerbated liver damage including inflammation, hepatobiliary damage, steatosis, and fibrosis, and increased liver enzymes in WT mice. Nevertheless, the role of ferroptosis in the context of alcohol and ACLF is yet to be explored.

The pleiotropic effects of alcohol in the context of end-stage liver disease go far beyond hepatocyte death. Cell death and specific types of cell death pathways are yet to be evaluated in other parenchymal and immune cells in the liver in ACLF. Alcohol also dampens the liver regenerative response [89], limiting hepatic progenitor cells to differentiate and replenish the hepatocyte pool, directly impairing gastrointestinal integrity and maturation and the function/activation of immune cells [90], contributing to systemic inflammation, immunopathology, and multi-organ dysfunction during ACLF. 

## 4. Hemolysis and Alcohol-Related Mortality 

### 4.1. Introduction to the Association between Alcohol and Hemolysis

Excessive alcohol intake causes half of all cases of liver cirrhosis worldwide and will be the leading etiology of advanced liver disease in the coming decades [91,92]. Despite its high prevalence and intensive research activities for over five decades, the underlying disease mechanisms are still poorly understood and so are typical laboratory characteristics [92]. For instance, in addition to the elevation of gamma-glutamyltransferase (GGT), heavy drinkers often show enlarged erythrocytes with an increased mean corpuscular volume (MCV) and elevated serum ferritin levels [93]. Moreover, in liver biopsies, about half of excessive drinkers show hepatic iron accumulation, which is almost evenly distributed in macrophages and hepatocytes [94,95,96]. 

It has been also known for a long time that alcohol consumption can lead to hemolytic anemia [97] most likely due to oxidative stress [98] and reduced phosphate levels [99]. In drinkers, morphological changes, such as echinocytes, stomatocytes, or spur cells, have been observed [97] and are prematurely removed by spleen macrophages [100,101]. So far, hemolysis has not been the focus of ALD studies for many years [102], except for a few case studies more than two decades ago describing morphological alterations of RBCs in alcoholic patients, such as stomatocytosis [99,103,104,105,106]. In addition, Zieve syndrome, a rare but severe form of hemolysis during heavy alcohol consumption, was first described in 1958 [102,107].

### 4.2. Heme Toxicity and Removal

Hemolysis is considered highly toxic because free heme can trigger diverse pro-oxidant and proinflammatory actions [108]. Therefore, erythrophagocytosis is essential for removing hemoglobin and its toxic metabolites. During erythrophagocytosis, heme oxygenase (HO) catalyzes the enzymatic degradation of heme and produces equimolar amounts of carbon monoxide (CO), biliverdin, and iron [108,109,110,111]. About 90% of iron is recycled from senescent erythrocytes that typically have a mean survival of 120 days [112,113,114]. In a coupled reaction, biliverdin is further converted into bilirubin (BR) via biliverdin reductase [115]. Of the two genetically distinct HO isoforms, the inducible HO isozyme HO-1 is expressed at low levels in most cells and tissues [111], whereas HO2 is constitutively expressed and mainly found in the brain and testis [116]. Of note, Nrf2 is a major upregulator of HO-1 that also orchestrates the transcriptional induction of various enzymes of the hepatic elimination and detoxification phases 0–3 [117]. Although macrophages, within the context of ALD pathology, have been intensively studied [118], the role of macrophages in recycling RBCs has received almost no attention. In addition to direct erythrophagocytosis, hemoglobin released from lysed RBCs is bound to either haptoglobin (Hp) or, after the release of toxic heme, hemopexin (Hpx), and rapidly internalized by macrophages through the hemoglobin–haptoglobin (Hb-Hp) complex-CD163 or the heme–hemopexin (Heme-Hx) complex-CD91 [119,120]. 

### 4.3. Novel Data from a 15-Year Prospective Survival Study in Heavy Drinkers: Mortality Is Primarily Defined by Hemolytic Anemia 

Preliminary data from a 15-year prospective ongoing survival study in heavy drinkers indicated that anemia is a major long-term predictor of death in these patients [94,121]. Patients enrolled in this study primarily presented for in-hospital alcohol detoxification for one week in the absence of specific liver-related symptoms. Further analysis demonstrated that patients with enlarged red blood cells (RBCs with high MCV) also had elevated ferritin, CD163, and LDH, and showed the highest mortality rate of about 30% during the mean follow-up for 4 years. Of note, this prognostically unfavorable high MCV sub-cohort had signs of enhanced erythropoiesis, thus fulfilling the criteria of so-called ineffective erythropoiesis. To our surprise, no deficiency of vitamin B12 or folic acid could be confirmed in these drinkers, suggesting that alcohol may be directly involved. These novel clinical findings focused on the elimination of senescent or damaged RBCs by phagocytic cells, a process termed erythrophagocytosis. 

### 4.4. Evidence for Enhanced Hemolysis and Erythrophagocytosis in Heavy Drinkers and a Chronic Ethanol Mice Model

In a large study cohort of heavy drinkers (n = 439), there is macroscopic evidence of optical hemolysis in 10% of all samples prior to alcohol detoxification, which decreased significantly to 5% after one week of alcohol withdrawal [122]. In silico tests of RBC fragility confirmed that RBCs from heavy drinkers are more fragile in response to the hemolytic agent phenylhydrazine (PHZ). Serum levels of hemolysis marker CD163, the scavenging receptor of the hemoglobin–haptoglobin complex, normalized after alcohol detoxification and continuously increased with increasing fibrosis stage. 

To learn more about erythrophagocytosis and alcohol, we also explored a chronic ethanol-feeding model [123]. Male wild type C57BL/6J mice were exposed to ethanol for 4 weeks, causing significant fatty liver and elevated transaminase levels. The sera of ethanol-fed animals showed a significantly higher prevalence of hemolysis. Enhanced hepatic heme degradation was confirmed by induction of HO-1 mRNA and significantly increased hepatic CD163 expression. 

### 4.5. Ethanol as a Direct Inducer of Hemolysis and Erythrophagocytosis

In another set of in vitro studies, human RBCs from healthy controls were directly exposed to increasing concentrations of ethanol. The data show that rather high ethanol levels >10%/1600 mM are required to cause direct and rapid hemolysis [122]. Next, immortalized human THP1 monocytes differentiated into macrophages were [124,125] exposed to RBCs that were pretreated with ethanol [122]. 

Only ethanol-pretreated RBCs were rapidly ingested by erythrophagocytosis, resulting in the accumulation of ingested RBCs inside THP1 cells (Figure 2A). Ethanol concentrations as low as 800 mM (5%) were required to prime RBCs for erythrophagocytosis, a process that can be blocked by the antioxidant N-acetylcysteine (NAC). Further mechanistic studies indicate that in addition to the CD163 receptor protein, Nrf2 (muclear factor erythroid 2-related factor 2), a major upregulator of *HO-1*, seems to be involved [117]. 

### 4.6. Evidence for RBC Ingestion (Efferocytosis) by Hepatocytes

As shown in Figure 2B, hepatocytes can also ingest oxidized or alcohol-pretreated RBCs [126] (see also Figure 2B). As hepatocytes are epithelial cells, this process should be better termed efferocytosis [127]. Some hepatocytes can ingest up to 20 RBCs. The direct uptake of RBCs by hepatocytes was also directly observed using a live video camera system. RBCs are taken up very rapidly within 10–15 min. While the exact receptor-mediated mechanism is still a matter of debate, our preliminary mechanistic studies indicate the role of *Nrf2* in efferocytosis.

### 4.7. Role of Hemolysis and Erythrophagocytosis in ALD

Figure 3 summarizes the multiple potential mechanisms of alcohol that ultimately cause hemolysis and enhanced erythrophagocytosis. [121]. These data provide a novel link between alcohol, the red blood cell compartment, and liver diseases. Consequently, Zieve syndrome, a long-time known but very rare form of severe hemolysis in heavy drinkers, could be just the top of the iceberg [107]. It is hoped that these novel insights will stimulate interventional studies to improve the treatment of severe complications of harmful alcohol consumption, such as liver cirrhosis or alcoholic hepatitis. Preliminary analysis in our large Heidelberg cohort confirms that signs of hemolysis are also tightly correlated with the presence of alcoholic hepatitis [128] and explain the rapid elimination of the specific ethanol biomarker, phosphatidylethanol [129]. Our findings could also provide a new molecular rationale for clinical studies using N-acetyl cysteine that decreased mortality in patients with alcoholic hepatitis [130]. They explain why levels of GOT/AST are typically elevated in drinkers, as they are directly derived from red blood cells [131]. 

It remains to be answered whether high alcohol levels prime RBCs for erythrophagocytosis or if direct hemolysis may be reached in heavy drinkers. The blood alcohol concentration (BAC) usually refers to the alcohol concentrations after complete distribution in all phases within the human organism. For example, in our cohort, the mean BAC was 1.0 permille. However, during consumption of various alcoholic beverages with alcohol concentrations typically ranging from 6% (beer) to high-percentage liquors up to 70%, the upper gastrointestinal tract and its associated vascular bed are exposed to much higher concentrations. Thus, higher alcohol levels have been reported for capillaries [132] and the portal venous system [133,134]. On the other side, our observations could provide a rationale for why liver damage in drinkers is delayed for several years and why a high percentage of beverages are more likely to cause alcoholic hepatitis.

It has yet to be studied how, e.g., RBCs obtain direct access to hepatocytes since they are normally separated by liver sinusoidal endothelial cells (LSECs) [135]. First, data indicate that LSECs are also able to ingest oxidized or alcohol-primed RBCs and are more sensitive to the toxic effects of heme degradation end products. Future studies should also explore the role of hemolysis in other liver diseases, such as metabolic dysfunction-associated steatotic liver disease (MASLD), Wilson’s disease, and many others.

## 5. Activation and Inactivation of HSCs in MetALD

Toxic liver fibrosis occurs in patients with chronic metabolic injury, alcohol consumption, and chronic hepatitis caused by viral infection. With the introduction of new therapies, the incidence of HBV and HCV fibrosis has declined [136], while metabolic dysfunction-associated steatohepatitis (MASH) and MetALD-induced fibrosis are on the rise [137]. MetALD remains a major risk factor for hepatic cirrhosis and HCC [136,137,138,139,140,141,142,143,144,145,146,147,148,149,150]. MetALD occurs in patients (BMI > 27) with MASLD, a spectrum of liver disease ranging from steatosis to non-alcoholic steatohepatitis (MASH) with fibrosis [151,152], and the intake of excess alcohol. 

The pathogenesis of MetALD is closely associated with metabolic syndrome, insulin resistance, and the development of hepatic steatosis. In comparison with MASH, excessive alcohol consumption strongly exacerbated liver injury [153]. Alcohol is metabolized in hepatocytes, leading to the production of toxic metabolites (acetaldehyde and acetate), damage, and apoptosis of hepatocytes, which release factors (including TGFβ, IL-8, IL-18, and others) that drive the recruitment of neutrophils into the damaged liver. In turn, neutrophils not only phagocytose apoptotic cells but also facilitate the flux of inflammatory monocytes into the liver. Monocytes release inflammatory and fibrogenic cytokines (IL-6, TNFα, IL-1β, TGF-β1) [150,154,155,156,157], triggering excessive de novo lipogenesis^2^ and the activation of HSCs into myofibroblasts. Myofibroblasts are absent in the normal liver but rapidly activate from liver resident mesenchymal cells into Collagen Type I and extracellular matrix (ECM) protein-producing myofibroblasts in response to fibrogenic stimuli to drive liver fibrosis. HSCs are believed to be the major source of myofibroblasts in MetALD and ALD livers [149,158,159,160]. 

### 5.1. The Role of Hepatic Stellate Cells in MetALD-Associated Liver Fibrosis

Under physiological conditions, quiescent HSCs (qHSCs) reside in the space of Disse, store vitamin A, and express neural (Lrat, NGFR1) [161,162] and lipogenic (PPARγ, Adipor1) markers [163]. In response to chronic alcohol or MetALD injury induced by intragastric alcohol feeding (Tsukamoto–French model), HSCs activate (aHSCs) and upregulate α-smooth muscle actin (α-SMA), vimentin, Col1a1, Col1a2, TIMP1, TGFβRI, Lox, LoxL1, and fibronectin. Although other cytokines (IL-6, leptin, CTGF, IL-17 [150,154,155,156,157]) also contribute to HSC activation, TGFβ1 is the major profibrogenic factor that drives Collagen Type I production in aHSCs [153] and triggers the activation of Smad2/3/4 signaling and the transcription of its target genes, PAI-1, Activin, and others [149]. The aHSC should be the primary target to treat MetALD fibrosis [149,158,159,160]. Moreover, the cessation of fibrogenic injury upon gradual weaning of mice from alcohol feeding often results in the regression of liver fibrosis and the inactivation of aHSCs [138,164]. Inactivated HSCs (iHSCs) stop producing Collagen Type I and acquire a quiescent-like phenotype (Figure 4) [165,166,167,168,169]. The regression of liver fibrosis in patients is well documented [152]. However, it remains unclear whether human aHSCs can be inactivated. 

Inactivation of HSCs (iHSCs): Using the Cre-Lox*P* genetic labeling of myofibroblasts, the fate of HSCs/myofibroblasts was elucidated during recovery from alcohol-induced liver fibrosis (Figure 4) and demonstrated that half of the myofibroblasts apoptose, while the other half of myofibroblasts escape apoptosis during the regression of liver fibrosis, downregulate fibrogenic genes, and acquire an inactivated phenotype, which is similar to but distinct from quiescent HSC (qHSCs). Inactivated HSCs (iHSCs) can retain some features of activated HSCs (aHSCs). iHSCs more rapidly reactivate into myofibroblasts in response to fibrogenic stimuli and more effectively contribute to liver fibrosis than qHSCs [138]. The inactivation of HSCs is associated with the re-expression of some lipogenic genes, PPAR-γ, Insig1, Bambi, NGFR1, and CREBP, but not the others (GFAP, Adipor1, Adpf) [168]. 

### 5.2. The Role of PPARγ in HSC Inactivation

The role of peroxisome proliferator-activated receptor gamma (PPARγ) in the regulation of alcohol-induced HSC activation in culture has been previously suggested. The ectopic overexpression of PPAR-γ results in the phenotypic reversal of activated HSCs to qHSC in culture [165,168,169] and the upregulation of adipogenic transcription factors causing a morphologic and biochemical reversal of activated HSCs to quiescent-like cells [170,171]. Findings in mice support these in vitro studies, demonstrating the importance of PPARγ for the maintenance of qHSCs and the inactivation of HSCs (iHSCs) during the regression of metabolic or alcohol-induced liver fibrosis [172]. We have demonstrated that PPARγ is differentially expressed in qHSCs, aHSCs, and iHSCs obtained from CCl_4_ and MetALD injured mice or after recovery (Figure 4). The re-expression of PPARγ is associated with HSC inactivation [138,172]. Consistently, iHSCs also upregulate the expression of quiescence-associated PPARγ target genes (Insig1, Egr1, and C/EBPd) and suppress the expression of profibrogenic PPARγ target genes (Cola1a, vimentin, and Cdk7) that are negatively regulated by PPARγ (Figure 4). 

### 5.3. Characterization of Mouse HSC Phenotypes

Original studies on the characterization of the distinct HSC phenotypes were performed using Rag2^−/−^γc^−/−^ mice adoptively transferred with mouse qHSCs, aHSCs, and iHSCs obtained from Collagen-α(I) mice: control mice, CCl_4_-injured mice, or mice that underwent the regression of liver fibrosis [138]. When mouse qHSCs, aHSCs, or iHSCs were transplanted into the livers of the Rag2^−/−^γc^−/−^ pups, all cells were equally engrafted. In turn, aHSCs and iHSCs exhibited growth advantage over qHSCs in CCl_4_-injured recipient Rag2^−/−^γc^−/−^ pups. aHSCs and iHSCs rapidly proliferated in response to CCl_4_, migrated to fibrotic lesions, and were fully integrated into liver architecture [138]. These data suggested that iHSCs maintain a “biological memory” of being activated. In accord, the repetitive administration of CCl_4_ into Collagen-α(I) mice revealed that mice previously exposed to CCl_4_ are more susceptible to recurrent injury and the development of liver fibrosis than the age-matched wild type littermates exposed to CCl_4_ at the time of the second insult [138]. In response to the second CCl4 injury, HSCs are more prompt to HSC activation, suggesting that iHSCs have some kind of biological memory of being activated [138].

### 5.4. Tools to Study Human HSC Inactivation

The inactivation of human HSCs cannot be mimicked in 2D cultures and requires a physiological environment. A new strategy has been developed to study the inactivation of human aHSCs. When mouse or human aHSCs were adoptively transplanted into the normal livers of immunodeficient Rag2^−/−^γc^−/−^ mice, they inactivate when placed into the physiological environment that does not provide fibrogenic stimuli, e.g., suppress the expression of fibrogenic genes and upregulate some quiescence-associated genes [138,172]. Human HSCs can be transplanted into the livers of one-day-old Rag2^−/−^γc^−/−^ mice. The resulting xenograft mice have human HSCs and retain functional human HSCs for almost 2.5 months [138]. Upon engraftment, when xenograft mice were subjected to toxic liver injury by the administration of carbon tetrachloride (CCl_4_), liver fibrosis was induced, and human HSCs underwent activation, thereby contributing to liver fibrosis in the xenograft mice. This suggests that phenotypic changes in human HSCs can be studied using this xenograft model [138]. 

### 5.5. Analysis of Epigenetic Marks That Regulate HSC Phenotypes

Chromatin immunoprecipitation linked to massively parallel deep sequencing (ChIP-Seq) has become a robust method to define the genomic binding locations of transcription factors, co-regulators, and specific histone modifications [172]. A histone post-translational modification enriched in *cis*-regulatory regions of transcriptionally active genes, histone H3 dimethylated at lysine 4 (H3K4me2), is a mark associated with promoters/enhancer activation [173]. Chromatin immunoprecipitation using antibodies against endogenous H3K4me2, the mark that correlates with cellular differentiation, identified promoters and enhancers of genes primed for expression. H3K4me3 immunoprecipitation is used to identify active promoters [174]. ChIP-Seq for the acetylated histone H4 lysine (H4K8ac) identifies the sites of transcriptional activation [175]. While H4K8ac has been shown as an important recruiter of p-Tefb (a complex composed of the kinase cdk9 and cyclin T1) [176,177], H3K27ac has been reported to be an excellent mark for identifying active enhancers [175,178]. Therefore, complimentary chromatin precipitation will be performed using the antibody of H3K27ac, which is often used to estimate cellular activation [175]. 

### 5.6. Transcriptional Regulation of HSC Phenotypes

Comparative analysis of the H3K4me2 pattern in HSCs (versus other cell types/tissues) has determined a strong enrichment of sequence motifs for ETS, IRF, Fox, NF1, and GATA regulatory elements in all HSCs [172], independent of activation state (Figure 4), suggesting that members of these transcription factor families may serve as the lineage-determining transcription factors responsible for HSC identity and cell type-specific responses. Enrichments for AP-1, TEAD, STAT3, and NF-κB motifs were observed in aHSCs [179]; while enrichments for ETS, IRF, and FOX motifs were associated with qHSC and iHSC phenotypes [172]. ETS1/2, GATA4/6, and IRF1/IRF2 were identified as putative mouse and human HSC lineage-specific TFs. Out of these transcription factors, ETS1 served as a master regulator of HSC lineage-determining TFs, which is responsible for the maintenance of quiescent-like phenotypes in HSCs. The deletion of ETS1 in HSCs exacerbates the development of toxic fibrosis in mice and results in the downregulation of *Ets1* target genes, *Nf1,* and *Pparγ* [172]. In turn, Gata6 was also implicated in the maintenance of quiescent HSC phenotypes. In vivo HSC-specific knockout of *Gata6* exacerbated the development of toxic liver fibrosis in mice [172]. In addition, Gata6 and Pparγ were implicated in driving the inactivation of mouse aHSCs. The deletion of Gata6 or Pparγ results in the attenuation of HSC inactivation, causing a defect of fibrosis resolution [172]. 

Despite extensive studies, there is no effective treatment for MetALD-associated liver fibrosis. Due to the complexity of MetALD pathogenesis, more than one signaling pathway might need to be targeted for successful anti-fibrotic therapy, which includes the suppression of excessive alcohol consumption, the treatment of metabolic syndrome, insulin resistance, inflammation, and fibrosis in patients with MetALD, and others. Meanwhile, reducing HSC activation may halt further progression of MetALD-associated liver fibrosis, while the underlaying etiology of the disease can be treated. Therefore, HSCs are the primary targets for anti-fibrotic therapy. Several pathways for targeting myofibroblasts/aHSCs have been suggested, including novel approaches, such as CART-mediated or immunotoxin-based ablation of senescent and/or activated HSCs [180,181].

## 6. Role of Hepatic Fatty Acid Binding-Protein 4 in Alcohol-Induced Hepatic Steatosis and Tumor Progression

A striking pathological event in the early stages of many liver diseases is increased hepatic lipid storage (hepatosteatosis), a feature readily detectable in ALD and MASLD [182,183,184]. An alcohol-associated steatotic liver develops in approximately 90% of chronic, heavy alcohol drinkers. Several mechanisms have been identified in the development and progression of hepatic steatosis, including the movement of lipids from other organs to the liver, the disruption of mitochondrial fatty acid β-oxidation, and sustained changes in the induction/expression of transcription factors associated with hepatic lipid metabolism, the cumulative effect of which is to elevate intracellular (hepatocyte) lipid content [185].

In the healthy liver, hepatic fat transport and lipid storage occur via several integrated pathways, including a central role for fatty acid binding proteins (FABPs) [186,187]. Functionally, FABPs bind free fatty acids (FFAs) with varying affinities, mechanisms, and ligand preferences [187,188]. To date, nine mammalian FABPs have been identified, with subtype expression being largely tissue specific [187,189]. For example, in the liver/hepatocytes, FABP1 is the predominant FABP detected, whereas FABP4 is localized to adipocytes and macrophages [189]. 

To date, the role of FABP4 in liver disease remains poorly defined. An analysis of liver tissue from a mouse model of chronic ethanol feeding (in which ethanol-induced hepatosteatosis occurs) reports that while FABP1 mRNA expression was unchanged, FABP4 mRNA expression was dramatically increased [190,191]. Western blot and immune–histochemical analysis identified increased hepatic FABP4 protein in ethanol-fed mice versus the control, which was hepatocyte specific [190]. Using tissue from ALD and HBV/HCV patients, a similarly robust increase in FABP4 mRNA was detected in ALD samples, concomitant with increased serum FABP4 levels (ELISA) compared to healthy subjects. However, analysis of liver samples and serum from HBV/HCV patients revealed that hepatic FABP4 mRNA expression and serum FABP4 protein levels were similar to those detected in healthy subjects [191]. 

Previous studies report that adipocyte-derived FABP4 can act as a paracrine/endocrine signaling entity to promote tumor progression in several different types of cancer arising adjacent to neighboring adipocytes (ovarian, prostate, and breast) [192,193]. Using in vitro cell culture models, exogenous recombinant human FABP4 (rhFABP4) is reported to stimulate HCC cell proliferation and migration via ERK-MAPK- and JNK1/2-dependent signaling pathways [191], and these data are consistent with other studies reporting that 25-hydroxycholesterol promotes HCC metastasis via increases in MMP9/FABP4 [194]. Conversely, other studies report a correlation between high HCC-FABP4 expression, recurrence-free survival in HCC patients with underlying HBV, and FABP4-dependent inhibition of cell proliferation in HBV^+^ cells in vitro [195], indicating that the underlying pathology impacts the role of FABP4 in hepatic tumor progression. 

Using cells transfected to overexpress cytochrome P450 2E1 (CYP2E1) or alcohol dehydrogenase (ADH), CYP2E1-dependent ethanol metabolism (but not ADH ethanol metabolism) leads to increased FABP4 mRNA expression and increased FABP4 protein detection in culture medium concomitant with increased intracellular neutral lipid accumulation [196]. Using a combination of pharmacological agents and molecular biology approaches, central roles for Sirtuin-1-forkhead box O1 (Sirt1-FOXO1) and AMP-activated protein kinaseα-sterol regulatory element binding protein-1c (AMPKα-SREBP-1c) signaling are reported to mediate CYP2E1-dependent ethanol metabolism in regulating hepatic FABP4 expression/secretion [196,197], studies that are supported by the identification of Sirt1 in regulating FABP4 expression in adipocytes [198].

Collectively, data from in vitro and in vivo model systems, analysis of human liver tissue/serum, and the literature identifying an endocrine role for adipocyte-derived FABP4 in diverse disease pathologies [199,200] demonstrate that ethanol-induced hepatosteatosis promotes de novo FABP4 synthesis and release from hepatocytes, and FABP4 stimulates HCC cell expansion (proliferation and migration) in vitro. These findings raise the intriguing possibility that, in addition to CYP2E1-dependent ethanol metabolism promoting hepatocyte transformation [201], this same pathway may promote FABP4 synthesis/release to promote transformed (cancerous) cell expansion. 

## 7. Clinically Relevant Models of ALD: Human Precision-Cut Liver Slices and Hepatic Organoids

Advances in drug development and outcomes in ALD have been significantly hampered by a lack of ex vivo models that recapitulate the complex clinical, histological, and molecular features of ALD. The imperative for developing robust clinically relevant models of ALD to pinpoint therapeutic targets and diagnostic/prognostic markers cannot be overstated, given its status as a pervasive global health concern characterized by a dearth of targeted therapies and alarmingly high mortality rates. Rodent models, while diverse, suffer from multiple limitations, such as incomplete fibrosis development and disparities in inflammation and immune responses to bacterial infection compared to humans [202], and, consequently, many targets identified in rodents fail during translation into human studies. Hepatotoxic, fibrogenic, and inflammatory responses in ALD are key drivers of disease progression, and animal models do not provide this disease-specific environment in response to ethanol. As such, they cannot accurately be utilized to identify therapeutically targetable molecular pathways, nor can they confer the opportunity to assess the efficacy of emerging drugs to these pathological events. These limitations, together with recent legislative shifts away from animal testing, exemplified by the FDA’s approval of alternatives [203], underscore the urgent need for improved human-relevant models that encompass the full complexity of ALD pathology, with particular emphasis on accelerated fibrogenesis and the inflammatory nature of the disease. In this context, emerging human-derived 3D culture methods offer promising avenues for modeling ALD, capitalizing on their functional and morphological advantages that capture intricate cell–cell interactions [204]. Liver organoids, generated from various hepatic cell types, despite their current functional immaturity, present promising avenues for ALD research. Additionally, liver slice culture, which enables the ex vivo culturing of thin tissue sections, retains cellular diversity and holds considerable potential for applications in the study of ALD. These advanced modeling techniques not only bridge the gap between basic research and clinical applications but also provide a more accurate representation of the complex pathophysiology of ALD. Ultimately, the development of improved models is crucial for advancing our understanding of ALD pathogenesis and accelerating the discovery of novel therapeutic interventions, leading to improved outcomes for patients suffering from this devastating disease.

### 7.1. Precision-Cut Liver Slices (PCLS): A Miniaturized Window into ALD Dynamics

The human PCLS model preserves the intricate architecture of the liver, maintaining both parenchymal and non-parenchymal cellular heterogeneity in culture. The unique aspect of PCLS lies in its preparation, which avoids the use of proteolytic enzymes, which are crucial for maintaining cell–cell interactions. First reported over 30 years ago, the PCLS preparation and experimental procedures have undergone significant refinement, supported by comprehensive guidelines for optimal practices [205]. This technique enables the monitoring of acute or chronic disease processes, offering valuable insights into the molecular mechanisms underlying ALD. The PCLS model also serves well as a platform for preclinical drug testing and effectively enables the exploration of acute and chronic scenarios, providing a bridge between preclinical and clinical studies not only in chronic liver diseases but also in liver cancers [206,207,208]. 

A recent study delves into the utility of human liver slices in mirroring the pathophysiological pathways associated with ALD [209], with a specific focus on key disease drivers of ALD, namely steatosis, hepatotoxicity, inflammation, and fibrogenesis. The PCLS retained viability in culture over 5 days together with the immune compartment and liver-specific histoarchitecture. In the context of recapitulating the molecular drivers of ALD, human PCLS obtained from human surgical tissue that would otherwise be discarded were acutely exposed to ethanol, fatty acids, and lipopolysaccharide, the characteristic trifecta of triggers observed in ALD. This resulted in increased lipid synthesis, hepatocyte death, and proinflammatory reactions in the slices, mirroring the hepatic response observed in ALD patients. In addition, the comprehensive molecular and histological characterization of PCLS revealed the development of hepatocyte megamitochondria in PCLS treated with ethanol [210,211], a distinctive and characteristic liver-specific histological marker of ALD. The study also explored gene expression, histology, and the release of M65 (epitope of cytokeratin-18), confirming the induction of hepatocyte death and revealing the intricate interplay of alcohol insults in replicating in vivo conditions. The study further uncovered proinflammatory responses in PCLS exposed to alcohol and bacterial insults. The release of prototypical proinflammatory cytokines (TNFα, IL-6, IL-8, and IL-1β) mirrored the inflammatory state observed in ALD patients. The profibrogenic activity of PCLS was confirmed via gene expression analysis and soluble markers. While mechanical cut induced wound-healing profibrogenic activity, alcohol exposure hindered the increase in slice thickness typically seen without alcohol exposure, indicating impaired wound-healing response to alcohol. This finding suggests a role for alcohol in modulating the regenerative capacity of the liver, a critical aspect in ALD development. Given the multifactorial nature of ALD, achieving therapeutic advancements and improved outcomes is likely to necessitate addressing several of these intrinsic pathways. The PCLS platform offers valuable insights into the primary drivers of disease and mortality in ALD and provides an opportunity to assess the efficacy of emerging therapeutics. While the PCLS model boasts numerous advantages, such as its ability to closely mimic clinical scenarios, it comes with limitations. These include inherent tissue donor heterogeneity and the acute nature of the model. However, this donor tissue variability can be thought of as enhancing the model’s representativeness of “real-life” clinical studies. Furthermore, to delve into the later stages of ALD, PCLS can be procured from explants obtained from patients with alcohol-associated cirrhosis undergoing liver transplantation, thereby expanding the scope of pathological mechanisms that can be investigated. 

### 7.2. Hepatic Organoids: Personalized Insights into ALD Pathophysiology

In parallel to PCLS, the advent of 3D culture methods has ushered in liver organoids as a promising avenue for studying hepatic physiology and pathophysiology. These organoids, derived from patient tissues or pluripotent stem cells, offer personalized models that preserve inter-individual features [212]. Suited for translational research and drug development, liver organoids also hold potential applications in regenerative medicine. Liver organoids, derived from various sources, including adult liver tissue, fetal liver tissue, bile, patient-derived tissue biopsies, or pluripotent stem cells, showcase versatility in modeling liver diseases, enabling the generation of personalized models that preserve inter-individual features. Despite facing challenges with inconsistent culture conditions hindering clinical application, a recent review by Hu et al. systematically compares recent methods, pinpointing controversial medium components to steer standardized liver organoid cultivation toward enhanced repeatability and efficacy [213].

The discovery of Lgr5-positive stem cells in the liver [214,215], coupled with advancements in culture protocols, has enabled the formation of chol-orgs (cholangiocyte-derived organoids) and hep-orgs (hepatocyte organoids) [216]. While chol-orgs closely mimic the biliary epithelium, hep-orgs show more advanced hepatic maturation, resembling primary human hepatocytes. Liver organoids derived from patient-specific iPSCs offer an exciting avenue for studying liver diseases in a translational setting, allowing the retention of genetic backgrounds, disease-causing mutations, and potential for precise genetic modifications. Various liver diseases, including ALD, have been successfully modeled using organoid culture systems [212]. A study in 2019 developed embryonic stem cell-derived hepatic organoids (hEHOs) and with the incorporation of human fetal liver mesenchymal cells showed ALD-associated pathophysiological changes, including oxidative stress, steatosis, inflammatory mediator release, and fibrosis [217]. Moreover, organoids from patients with cirrhosis demonstrate the ability to mimic ductular reactions in alcoholic hepatitis, providing a unique platform to study and assess functional roles. The proinflammatory profile expressed by these organoids further indicates their potential in mimicking disease conditions [218]. Overall, organoids represent a promising tool for advancing our understanding of liver biology and disease pathology, despite certain limitations, including reproducibility, scalability, and costs.

## 8. Summary and Conclusions

Alcohol promotes epigenetic changes in multiple cell types in the liver, which contribute to alcohol-induced liver injury, inflammation, steatosis, and fibrosis, as well as alcohol-associated HCC. Epigenetic enzymes could serve as a future target for various aspects of alcohol-associated liver disease.End-stage liver disease involves different types of cell death, including apoptosis, necroptosis, pyroptosis, and ferroptosis, with a predominant pathway depending on the etiological agent causing the disease progression. Novel therapeutic approaches to prevent excessive cell death during the late stages of chronic liver disease are needed. Moreover, alcohol impairs liver regenerative responses and directly impairs gastrointestinal integrity and the function/activation of immune cells, contributing to systemic inflammation and multi-organ dysfunction during ACLF.Novel data indicate that alcohol triggers hemolysis and also primes RBCs for enhanced erythrophagocytosis by macrophages and hepatocytes; thereby, RBCs are continuously primed for elimination and increased heme turnover. Toxic end products of hemolysis and end-stage liver disease, such as bilirubin or heme, can further enhance erythrophagocytosis, potentially allowing the initiation of a vicious cycle. These novel observations are likely to contribute to hepatic iron overload in ALD and end-stage ALD and alcoholic hepatitis.The inactivation of aHSCs can provide a novel strategy for the suppression of liver fibrosis. However, the pathways and modulators of HSC activation will need further characterization. One of the potent strategies is based on the manipulation of aHSC epigenetic marks that can lead to HSC inactivation.Ethanol-induced liver steatosis promotes de novo FABP4 synthesis and release from hepatocytes, and FABP4 stimulates HCC cell expansion (proliferation and migration) in vitro. These findings raise the possibility that in addition to CYP2E1-dependent ethanol metabolism promoting hepatocyte transformation; this same pathway may promote FABP4 synthesis/release to promote transformed (cancerous) cell expansion.The integration of human PCLS and hepatic organoids represents a significant advancement in modeling ALD, offering a representation of the human disease. The combined use of precision-cut liver slices and organoids in preclinical research holds promise for unlocking new insights into ALD pathogenesis, advancing the development of targeted therapeutic interventions, and, overarchingly, improving outcomes in ALD.

## Figures and Tables

**Figure 1 biomolecules-14-00404-f001:**
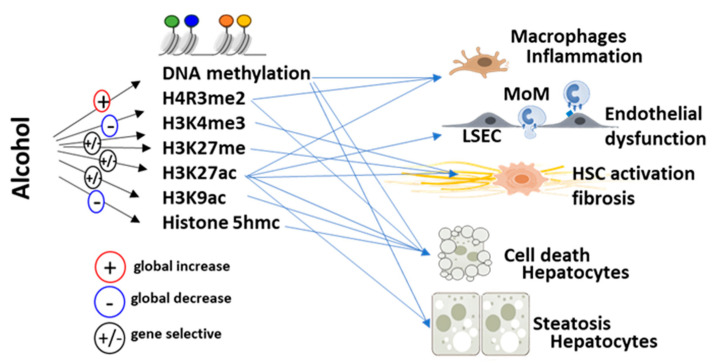
Schematic of major alcohol-induced histone modifications. Major modifications that are affected by alcohol are histone acetylation and DNA and histone methylation. Alcohol promotes a global increase or decrease in several modifications while affecting others in a gene-specific manner (**left**). These modifications were reported to contribute to alcohol-induced liver inflammation, endothelial dysfunction, hepatic stellate cell activation, liver injury, and hepatocyte steatosis (**right**).

**Figure 2 biomolecules-14-00404-f002:**
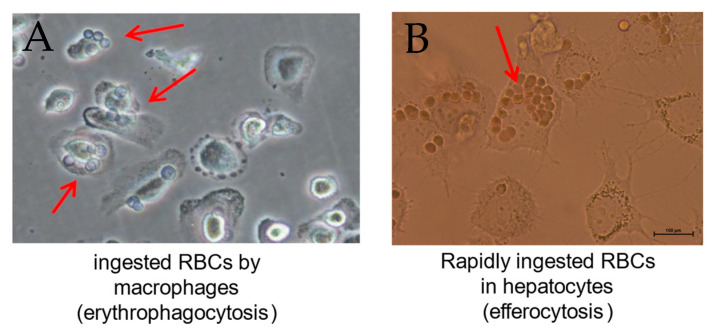
Erythrophagocytosis and hepatocyte-mediated efferocytosis of red blood cells pretreated with ethanol. Human RBCs were pretreated for 24 h with 800 mM ethanol and then cocultured with (**A**) human THP-1 macrophages or (**B**) Huh7 hepatoma cells for a further 24 h. As demonstrated in both figures, multiple RBCs can be ingested (arrows). They surround the nucleus, confirming complete internalization B, red arrow). Pictures are directly taken from cell culture dishes using inverted live microscopy.

**Figure 3 biomolecules-14-00404-f003:**
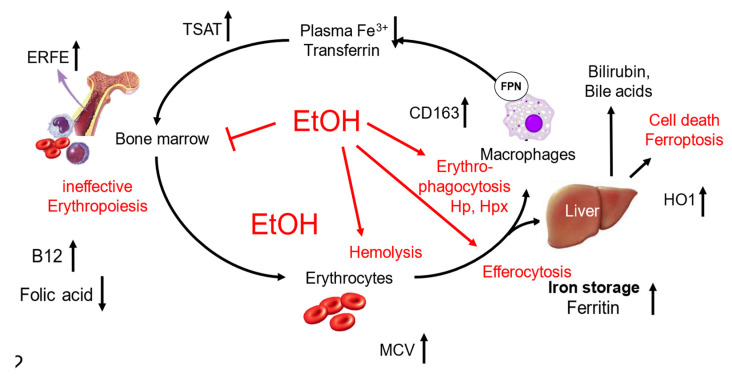
Alcohol and its effects on the red blood cell turnover and laboratory parameters. Hemolytic anemia is one of the major factors associated with long-term mortality in heavy drinkers. Generally, RBC turnover is enhanced. Ethanol interferes with RBC turnover at three major sides. First, it can block hematopoiesis, second, it increases RBC fragility, and third, it primes RBCs for both erythrophagocytosis and efferocytosis. Major consequences are elevated ferritin and MCV, suppressed transferrin, and low RBC count. Of note, B12 is typically elevated in ALD patients, and only folic acid levels are decreased or in the lower normal range. There are first indications that the ingestion of RBCs causes subsequent cell death, most likely due to the release of toxic iron and other compounds. The detailed role of RBC efferocytosis by hepatocytes and their interaction with endothelial cells needs to be addressed in future studies. Abbreviations: ERFE; erythroferrone, EtOH, ethanol; HO1, hemoxygenase-1; MCV, mean corpuscular volume; TSAT, transferrin saturation.

**Figure 4 biomolecules-14-00404-f004:**
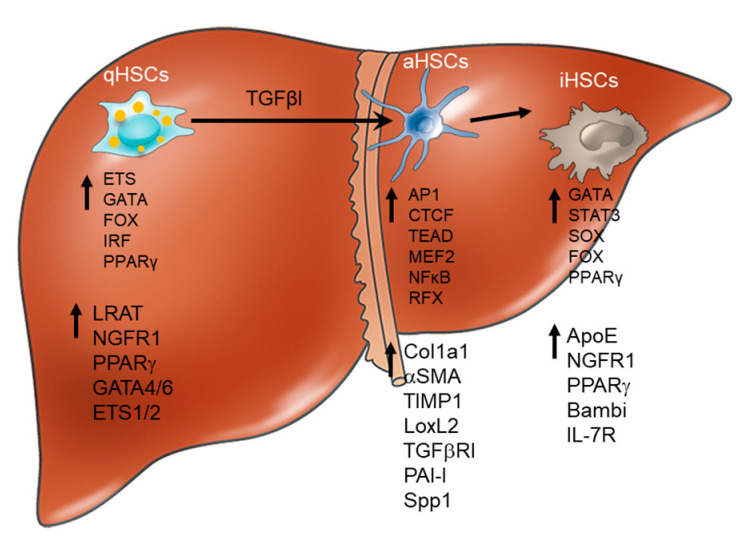
**Phenotypic changes in mouse HSCs**. Upon development and regression of liver fibrosis. Quiescent HSCs (qHSCs) undergo activation into hepatic myofibroblasts (aHSCs) in response to chronic liver injury. Upon cessation of etiological injury, aHSCs can apoptose or inactivate (iHSCs) in to quiescent-like HSCs. Characteristic genes and regulatory TFs specific for each HSC phenotype. HSC phenotypes are associated with upregulation or downregulation of specific genes: qHSCs express lipogenic and neural markers, but in response to chronic liver injury downregulate these genes and upregulate markers of fibrogenic myofibroblasts. Regulation of HSCs is mediated by a set of specific transcription factors that are lineage-specific and activation-specific. The key transcription factors critical for each phenotype are listed. 
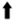
 denotes up-regulation.

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
