# Peer review of "Alcohol-Associated Liver Disease Outcomes: Critical Mechanisms of Liver Injury Progression"

_biomolecules, 2024, doi:10.3390/biom14040404_

Round 1

Reviewer 1 Report

Comments and Suggestions for Authors

In the manuscript entitled “Title Alcohol-Associated Liver Disease Outcomes: Critical Mechanisms of Liver Injury Progression” Osna N et al have reviewed the current understanding of ALD and its mechanisms leading to liver injury progression. The review covers major mechanistic insights in the field and is of interest to the study of ALD and associated liver abnormalities. Insights into the role of hemolysis and FABP in ALD are described well. A few suggestions would add more value to this review:

1.      Section 2.1. Alcohol-Induced Hepatocyte Epigenetic Changes: The authors review the role of alcohol in modulating epigenetics. The role of alcohol in modulating acetylation of mitochondrial proteins and its regulation by deacetylases should also be discussed since this is a mechanism that contributes to alcohol-dependent lipid dysregulation.

Author Response

We thank you and the reviewers for their helpful comments for revision. We have point-by-point addressed all the concerns of the reviewers’ and have tracked the changes in the submitted manuscript. Below, we provide a point-to-point response to the reviewers’ comments.

Reviewer 1

Comment: Section 2.1. Alcohol-Induced Hepatocyte Epigenetic Changes: The authors review the role of alcohol in modulating epigenetics. The role of alcohol in modulating acetylation of mitochondrial proteins and its regulation by deacetylases should also be discussed since this is a mechanism that contributes to alcohol-dependent lipid dysregulation.

Response: We agree that alcohol induces acetylation of the mitochondrial proteins which in turn impacts alcohol induced lipid dysregulation. However, this mechanism is beyond the scope of epigenetic regulation, which generally refers to histone modifications.

Reviewer 2 Report

Comments and Suggestions for Authors

Overall, this review manuscript provided a comprehensive overview of alcohol related liver diseases, which is very useful for new and active researchers in ALD. However, multiple sections of the manuscript lack conciseness with big paragraphs but few mechanistic details about ALD.  The manuscript indicates Fig.1B in various places (line 508,579,489), but the present fig doesn't match the concepts in those portions. Below are a few minor comments.

1)      Figures should be explained in detail (elaborate legends).

2)      It would be more helpful to the readers if the manuscript included schematics and some technical details on PCLS and hepatic organoids.

3)      I find it strange that only FABP was selected for the Alcohol induced hepatic steatosis section of the paper. In general, they could have discussed the steatosis effect of alcohol and then discussed FABP's importance.

Comments on the Quality of English Language

NA

Author Response

We thank the reviewer for their fruitful comments. We revised the manuscript according to their recommendations (please find the detailed responses below):

Comment 1: Overall, this review manuscript provided a comprehensive overview of alcohol related liver diseases, which is very useful for new and active researchers in ALD. However, multiple sections of the manuscript lack conciseness with big paragraphs but few mechanistic details about ALD.  The manuscript indicates Fig.1B in various places (line 508,579,489), but the present fig doesn't match the concepts in those portions. Below are a few minor comments.

Response: Corrected

      Comment 2: Figures should be explained in detail (elaborate legends).

      Response:  Corrected

      Comment 3: It would be more helpful to the readers if the manuscript included schematics and some technical details on PCLS and hepatic organoids.

      Response: For the PCLS – The methods have been published several times and in much detail. To aid the reader in selecting the best methodology we have now inserted a sentence and reference  ‘First reported over 30 years ago, the PCLS preparation and experimental procedures have undergone significant refinement, supported by comprehensive guidelines for optimal practices (doi: 10.3390/ijms22137137, PMID: 34281187).

      For the Organoids – Again we have steered the reader to the best challenges and successes in Liver Organoid Technology and inserted at Line 744 –‘Despite facing challenges with inconsistent culture conditions hindering clinical application, a recent review by Hu et al. systematically compares recent methods, pinpointing controversial medium components to steer standardized liver organoid cultivation towards enhanced repeatability and efficacy (doi: 10.1186/s13578-023-01136-x, PMID: 37915043)’.

Comment 4:   I find it strange that only FABP was selected for the Alcohol induced hepatic steatosis section of the paper. In general, they could have discussed the steatosis effect of alcohol and then discussed FABP's importance.

Response: We thank the reviewer for their feedback. The manuscript has been revised to include background regarding the role of ethanol in initiating fat accumulation in hepatocytes (hepatosteatosis) as a rationale for studying mechanisms that regulate (and become dysregulated during) lipid transport and storage.

Reviewer 3 Report

Comments and Suggestions for Authors

Section 2: Epigenetics in ALD Progression.  Please include a schematic showing key summary points from this section. 

Section 3: Acute-on-Chronic Liver Failure: See section 3.2 – second paragraph, line 235, “Our recent work presented at The Liver Meeting from the AASLD….” Please revision this introductory section and remove use of terms like “our” and “we” here and throughout the entire article. 

Section 4: Hemolysis and Alcohol-Related Mortality: This section of the article requires significant revision, editing, and streamlining. First, section is too long in relation to other sections of the article. Second, authors should reconsider organization and structure of this section – e.g., begin with more clinical and epidemiological information, ending with more basic science information.  It is appropriate to refer to the chronic + binge model of alcohol feeding in mice to the “NIAAA” model?  Does NIAAA specifically endorse this model? Line 345, Revise use of “B6” mice to Wild-type C57BL/6 (strain J or N) mice, and also include sex of mice.  Reconsider including Figure 1 – images are not high quality and data are not included in other sections.  Overall, this section “reads” like I am listening to an investigator giving a talk, especially Section 4.5. Lastly, final paragraphs, Lines 424-442, require significant editing/revising to improve quality.

Section 5: Activation and Inactivation of HSCs in MetALD:  First, there is little to no scientific results from MetALD presented in this section; only chemical toxicant models (CCL4).  Therefore, please reconsider stating that this section is on MetALD. Second, revise last sentence, Line 465, do not begin sentence with, “This mini-review…”  Second, do not use terms like “we” and “our”. Third, the authors refer to figures that are not included – Figures 1A and B.  Please have the writer fix this mistake, and check section for any “cut-and-paste” errors. Lastly, the authors might consider moving Section 5 to follow Section 2, which is also focused on epigenetics. There is redundancy between this sections. Combining and streamlining sections 2 and 5 would likely improve readability.

Section 6: FABP4 – This section is nicely written and requires no revision.

Section 7: Clinically relevant models: Again, reconsider use of terms “we” and “our”. Could the authors be more precise in some of their findings, e.g., what is meant by, “alcohol exposure significantly impacted slice thickness…”  Did it increase or decrease or cause some other anatomical change?

Section 8: Summary and Conclusions.  This is a nice section; however, it is a bit long. Some editing and streamlining is suggested.

One final comment – The authors present a fair bit of their data (animal and clinical), and some that may be unpublished. As such, this reviewer wonders whether IACUC and IRB approvals should also be included, for example. Please consult with journal editors on this point.

Thank you for an enjoyable and highly educational review on ALD.

Comments on the Quality of English Language

English quality is fine.  Paper just need minor editing to correct typos and a few missing words.

Author Response

We thank the reviewer for their fruitful comments. We revised the manuscript according to their recommendations (please find the detailed responses below):

Comment 1:Section 2: Epigenetics in ALD Progression.  Please include a schematic showing key summary points from this section. 

Response: We now added a schematic of key points (new Figure 1)

Comment 2: Section 3: Acute-on-Chronic Liver Failure: See section 3.2 – second paragraph, line 235, “Our recent work presented at The Liver Meeting from the AASLD….” Please revision this introductory section and remove use of terms like “our” and “we” here and throughout the entire article. 

Response: Corrected

Comment 3: Section 4: Hemolysis and Alcohol-Related Mortality: This section of the article requires significant revision, editing, and streamlining. First, section is too long in relation to other sections of the article.

Response: We have significantly shortened the whole section

Comment 4: Second, authors should reconsider organization and structure of this section – e.g., begin with more clinical and epidemiological information, ending with more basic science information. 

Response: As proposed, we have added to further sections with subheadings for more clarity, structure and readability

Comment 5: It is appropriate to refer to the chronic + binge model of alcohol feeding in mice to the “NIAAA” model?  Does NIAAA specifically endorse this model?

Response: To avoid any confusion, we now avoid the term NIAAA model.

Comment 6:Line 345, Revise use of “B6” mice to Wild-type C57BL/6 (strain J or N) mice, and also include sex of mice.  

Response: Corrected

Comment 7: Reconsider including Figure 1 – images are not high quality and data are not included in other sections.  

We have added Figure 1.

Comment 8: Overall, this section “reads” like I am listening to an investigator giving a talk, especially Section 4.5. Lastly, final paragraphs, Lines 424-442, require significant editing/revising to improve quality.

Response: Section has been revised and shortened

Comment 9:Section 5: Activation and Inactivation of HSCs in MetALD:  First, there is little to no scientific results from MetALD presented in this section; only chemical toxicant models (CCL4).  Therefore, please reconsider stating that this section is on MetALD. Second, revise last sentence, Line 465, do not begin sentence with, “This mini-review…”  Second, do not use terms like “we” and “our”. Third, the authors refer to figures that are not included – Figures 1A and B.  Please have the writer fix this mistake, and check section for any “cut-and-paste” errors. Lastly, the authors might consider moving Section 5 to follow Section 2, which is also focused on epigenetics. There is redundancy between this sections. Combining and streamlining sections 2 and 5 would likely improve readability.

Response: Everything is corrected. Sections 5.5 on Epigenetic regulation of HSC phenotypes is deleted. 

Comment 10:  Section 6: FABP4 – This section is nicely written and requires no revision.

Response: Thank you for this comment. However, this section has been revised based on Comment #4 of Reviewer 2

Comment 11: Section 7: Clinically relevant models: Again, reconsider use of terms “we” and “our”. Could the authors be more precise in some of their findings, e.g., what is meant by, “alcohol exposure significantly impacted slice thickness…”  Did it increase or decrease or cause some other anatomical change?

Response: This has now been addressed.

Comment 12: Section 8: Summary and Conclusions.  This is a nice section; however, it is a bit long. Some editing and streamlining is suggested.

Response: Corrected

Comment 13:One final comment – The authors present a fair bit of their data (animal and clinical), and some that may be unpublished. As such, this reviewer wonders whether IACUC and IRB approvals should also be included, for example.

Response: There are only references to unpublished data, which were briefly mentioned. No original data have been presented. All aforementioned studies were approved by Institutional Committees, but the inclusion of IACUC and IRB are not required for the review paper.

Round 2

Reviewer 3 Report

Comments and Suggestions for Authors

No comments

Comments on the Quality of English Language

Please make sure that grammar and formatting issues are corrected.